# Use of Lactoperoxidase Inhibitory Effects to Extend the Shelf Life of Meat and Meat Products

**DOI:** 10.3390/microorganisms12051010

**Published:** 2024-05-17

**Authors:** Filip Beňo, Adéla Velková, Filip Hruška, Rudolf Ševčík

**Affiliations:** Department of Food Preservation, Faculty of Food and Biochemical Technology, University of Chemistry and Technology Prague, Technická 5, 166 28 Prague, Czech Republic

**Keywords:** lactoperoxidase, meat products, *Listeria innocua*, *Staphylococcus saprophyticus*, *Pseudomonas fluorescens*, TBARS

## Abstract

Lactoperoxidase (LP) is an important enzyme of the salivary and mammary glands. It has been proven to increase the shelf life of raw milk by inhibiting the growth of bacteria, especially *Listeria monocytogenes*, *Escherichia coli*, *Staphylococcus aureus*, and *Pseudomonas* spp. The aim of this work was to verify the use of LP to extend the shelf life of meat products. In vitro experiments showed inhibitory effects on the selected bacteria (*Listeria innocua* (ATCC 33090), *Staphylococcus saprophyticus* (CP054440.1), and *Pseudomonas fluorescens* (ATCC 13525) due to a prolongation of the lag phase of growth curves. A lower increase in viable counts (*p* < 0.05) was also found by testing pork cubes’ surface treated with LP solution (5%) + *L. innocua* and stored for 7 days at 15 °C. LP has also been studied at concentrations of 0.25 and 0.50% in meat products (pork ham and pâté) during refrigerated storage (4 °C for 28 days). Lower viable counts were observed throughout the storage experiment, especially for 0.50% LP (*p* < 0.05). Meat products containing LP also showed lower levels of oxidation (MAD) (*p* < 0.05). According to these results, LP could extend the shelf life of a wider range of products.

## 1. Introduction

Currently, there are many licensed chemical preservatives (especially nitrites) that are used to extend the shelf life of meat products. However, we can increasingly find claims that question these preservatives due to their possible negative effects on human health [1,2,3].

Lactoperoxidase (LP) is an enzyme that belongs to the peroxidase group. It is a group of widespread natural enzymes that are found in the secretions of the mammary (colostrum and milk), salivary and lacrimal glands [4]. The main synthesis of LP occurs in the mammary glands, where it then performs a protective function against pathogenic microorganisms [5]. The antimicrobial activity of LP is based on the enzyme activation and formation of hypothocyanite ions [OSCN]^−^ as reagents with bacterial membranes [6] affecting the metabolic enzymes of microorganisms [7] and cell function, e.g., membranes’ integrity and the transport systems of membranes (the release of polypeptides, potassium ions, and amino acids and cellular uptake of purines, pyrimidines, glucose, and amino acids are inhibited) [8,9]. Oxidation of sulfhydryl (SH) groups of enzymes and proteins is considered to be crucial. If a high level of protein sulfhydryl oxidation occurs, the amino acid residues histidine, tryptophan, and tyrosine are modified [10]. The antibacterial effect of LP against different groups of microorganisms shows varying degrees of sensitivity, potentially having a bactericidal or bacteriostatic effect depending on factors such as the type of microorganism, the type of electron donor in membrane proteins, pH, temperature, incubation time, and concentration of microorganisms [11].

The activity and inhibitory action of LP is different for various types of bacteria such as Gram-negative and Gram-positive bacteria [8]. In most studies, a positive effect on inhibiting the growth or reducing the number of foodborne pathogens is presented. In the case of Gram-negative bacterial foodborne pathogens such as *Salmonella* spp., *Pseudomonas* spp., *Escherichia coli*, *Shigella* spp., and *Klebsiella* spp., the LP enzyme caused either an inhibitory or a lethal effect. The killing of bacteria does not only consist of direct killing due to the effect of LP on the life-sustaining metabolic processes of the bacteria. The death and subsequent lysis of the bacterial cell causes the release of some nutrients into the environment preventing other bacteria from absorbing nutrients that leads to a reduction in their numbers or growth inhibition [12,13]. It is assumed that, in general, the cell wall of Gram-positive bacteria shows higher resistance compared to Gram-negative bacteria. For example, smooth strains of *Salmonella typhimurium* have a complete polysaccharide sequence and are thus less susceptible to LP than rough strains [8]. Regarding the LP effect on Gram-positive foodborne pathogen bacteria (*Listeria monocytogenes*, *Staphylococcus aureus*, *Bacillus* spp., *Clostridium perfingens*, etc.), the LP has an inhibitory effect on these bacteria [13,14,15,16].

Another group of Gram-positive bacteria we can mention are lactic acid bacteria (LAB). For example, streptococci are reported to be able to repair damage to the bacterial wall or at least partially degrade oxidation products produced by LP [8]. This statement is then consistent with the fact that Gram-positive bacteria are generally more resistant to LP [13,14,15,16]. In previous work focused on cheese curdling time and ripening, it was proved that there was no effect of acidity development in milk with an activated LP system. This suggests a high resistance of starter cultures used for cheese manufacture [8]. One review article also mentions that an adult toothpaste containing LP did not have any antibacterial effects on *Streptococcus mutans* and lactobacilli over a 4-week period [17]. Since LAB are often used for their beneficial properties, for example for fermentation of products, then LP could act as an additional preservative that destroys or inhibits unwanted microorganisms and pathogens but does not affect LAB. Another advantage of using LP as a preservative is the GRAS (generally recognized as safe) categorization, leading to numerous expert publications on its use as a natural food preservative instead of chemical preservatives [16,18].

LP shows activity in the pH range from 4 to 11 and maximum activity is reached at pH 6. However, it is also resistant to acidic pH values up to 3. In general, LP is considered to be an enzyme with overall high heat stability and is reported to be one of the most heat-stable enzymes found in milk; its thermal inactivation has been accepted as a criterion for the high temperature pasteurization of milk. Research on its stability is varied and shows up some differences. Short-term pasteurization at 74 °C only inactivates LP and retains sufficient enzyme activity [19]. Inhibited LP activity was also detected when cow’s milk was pasteurized for 30 min and 15 s at 63 °C and 72 °C, respectively, but destruction occurred when the temperature reached 80 °C for 2.5 s [20]. Complete inactivation of the LP enzyme was also found during pasteurization of cow’s milk at 78 °C for 15 s [21]. Other results are reported by a study where a still active LP system was found after pasteurization of cow’s milk at 72 °C for 15 s, which is still able to maintain the quality of milk inoculated with the bacteria *Pseudomonas aeruginosa*, *Staphylococcus aureus*, and *Streptococcus thermophilus* [21]. It is also necessary to mention the increased sensitivity of LP to high temperatures at a low pH value [7,22].

Due to the thermal stability of LP, there is potential for this enzyme to be used for meat production and to extend the shelf life of meat products. Meat products are often heat-treated and in the case of the Czech Republic, meat products of the “Heat-treated” group according to Decree No. 69/2016 Coll. must be heat-treated in the core of the product for 10 min at a temperature of 70 °C [23]. At present, as far as the authors know, there are not many publications that have examined the potential of LP to extend the shelf life of meat products. For illustration, the study on the effect of LP on microbial contamination of the surface of beef cubes inoculated with approximately 10^4^ CFU/cm^2^ of *L. monocytogenes* and incubated at different temperatures (37 and 12 °C) revealed that LP was found to have a greater effect at lower non-permissive temperatures for rapid bacterial growth with strong growth inhibition [9]. The temperature dependence of the inhibitory effect of LP was also confirmed in a study by monitoring the growth of *E. coli* O157:H7, *L. monocytogenes* L45, *S. aureus* R37 in broth (37 °C), and ground beef (0, 6, and 12 °C). The inhibitory effect was confirmed and, in addition, *L. monocytogenes* was found to be more sensitive to LP compared to the other bacteria [1].

Considering the possible use of LP for the preservation of meat products, the inhibitory effect of LP on *L. innocua*, *S. saprophyticus*, and *P. fluorescens* in liquid broth (different incubation temperatures) and on the surface of pork cubes as well as directly in meat products (pork ham and pâté) was investigated in this experiment.

## 2. Materials and Methods

### 2.1. Lactoperoxidase

In this work, a mixture of active enzyme and substrate (maltodextrin) based on the lactoperoxidase system HYDRO LP (Hubka-Petrášek a vnuci, Ltd., Bavoryně, Czech Republic) was used. The dose of this mixture in products ranges from 4 to 6 g per kg of product. This product (enzyme LP) was obtained by extraction from cow’s milk.

### 2.2. Testing of LP in Liquid Broth

The inhibitory effect of LP was tested in vitro on the growth of bacteria *L. innocua* (ATCC 33090) in liquid broth of TSB (tryptic soy broth) (Merck KGaA; Darmstadt, Germany), where (CP054440.1) (isolated from pork meat) in liquid broth of BHI (brain heart infusion) (Merck KGaA; Darmstadt, Germany), and *P. fluorescens* (ATCC 13525) in TSB broth. Non-pathogenic *L. innocua* was selected as a replacement for pathogenic *L. monocytogenes*. In this part of the experiment, the development and changes in bacterial growth curves (time dependence of the optical density of microorganisms) were monitored using a laboratory bioreactor Biosan RTS-8 (Biosan, Riga, Latvia) according to [24] with modifications. The measurement is performed at a wavelength of 850 ± 15 nm to eliminate the effect of the color of the culture medium and the plastic tube used.

Measurements were carried out in parallel in sterile conical tubes (LaboServ, Brno, Czech Republic) with a volume of 50 mL into which 14 mL of TSB broth + 1 mL of *L. innocua* inoculum (3.1 × 10^7^ CFU/mL) + 1 mL of 1% LP (in sterile distilled water) were transferred. The tubes thus prepared were placed in a bioreactor and cultured at 15, 20, 25, and 40 °C with a speed of 2000 rpm. The same procedure was carried out with bacteria *P. fluorescens* (2.4 × 10^8^ CFU/mL). To prevent oxygen access, the measurement process of *S. saprophyticus* (1.3 × 10^7^ CFU/mL) was modified—rpm 2000 only before the measurement itself, not continuously. Similarly, control samples containing only broth (15 mL) + inoculum (1 mL) of chosen bacteria *L. innocua*, *S. saprophyticus* or *P. fluorescens* at the same inoculum concentrations were also prepared in parallel.

### 2.3. Testing of LP on Pork Cubes

The inhibitory effect ex vivo of LP on the surface of pork (shoulder) cubes (2 × 2 × 2 cm), which were divided into 3 groups—control cubes soaked only in distilled water, cubes soaked in the TSB broth containing *L. innocua* (2.5 × 10^6^ CFU/mL), and cubes soaked in *L. innocua* (2.5 × 10^6^ CFU/mL) and then LP solution (5 g of LP in 100 mL distilled water). Soaking the cubes took 1 min and took place under sterile conditions in flow box (Schoeller Instruments, Ltd., Prague, Czech Republic), as did cutting of the meat into cubes. All solutions were tempered to a temperature of 20 °C before soaking. After soaking, the cubes were kept in drain excess fluids, placed into sterile conical tubes and transferred to a thermostat POL-EKO ST2B40 (POL-EKO, Wodzisław Śląski, Poland), and left to cultivate at 15 °C for 1, 4, and 7 days. This was followed by microbial analyses; all microbiological analyses were performed in triplicate (three cubes from each treatment). Inoculations were performed on PCA (plate count agar) (Penta, Prague, Czech Republic) for a TVC (total viable count) determination and culture was carried out at 30 °C for 3 days.

### 2.4. Testing of LP in Meat Products

To test the effect of LP in a meat product on its microbiological stability, heat-treated meat products (pork ham and pâté) were chosen and produced under laboratory conditions. Pork leg meat (purchased from the market) was used to produce restructured ham as well as the remaining raw materials for the production of pâté.

A total of three hams were produced and the composition was as follows: pork leg meat (330.0 ± 1.0 g) + curing salt (5.0 g) + water (66.0 g) + 0.25%, respectively 0.50% LP. The composition of pâtés was as follows: pork liver (380.5 ± 1.0 g) + pork lard (400.0 ± 1.0 g) + pork flank (760.5 ± 1.0 g) + pork broth (165.5 ± 0.5 g) + curing salt (31.0 g) + dried onion (70.5 ± 0.1 g) + garlic. (15.1 ± 0.1 g) + spice (12.1 ± 0.1 g) + 0.25%, respectively 0.50% LP.

The pork cubes (1 × 1 × 1 cm) together with the curing salt were mixed at first for 15 min, and then ½ the volume of water was added. After another 15 min, the rest of the water was added and mixing continued for a further 55 min. The mixture was transferred to polyamide bags and plastic cylinders mold before heat treatment. The pâtés were made using a Thermomix TM5 (VORWERK, Wuppertal, Germany). First, the liver was crushed with curing salt to mousse, then lard, boiled flank and broth were added and mixed to a soft consistency. Finally, onions, garlic, spices, and LP were added. The finished mixture was filled into jars (content about 75 g) and heat-treated. Subsequently, the hams and pâtés were cooked in a Memmert WB22 water bath (Memmert GmbH, Schwabach, Germany) to the Czech legislative requirements of 70 °C in the core of the product for 10 min. The finished meat products were then cut into 1 cm slices, wrapped in a polyamide bag, placed in a POL-EKO ST2B40 (POL-EKO, Wodzisław Śląski, Poland), and stored at 4 °C with constant illumination by LED lamps. The meat products were then subjected to a storage experiment and microbiological analyses were carried out on days 1, 7, 14, 21, and 28.

### 2.5. Microbial Analysis of Meat Products

The sample (10 g) was weighed in a polymer bag along with 90 mL of saline and homogenized in a homogenizer MIXW 1002 (MIXWEL^®^; Bruz, France) at the highest intensity for 3 min. The saline solution was prepared by dissolving 8.5 g of NaCl (Penta; Prague, Czech Republic) and 1 g of peptone (Merck KGaA; Darmstadt, Germany) in 1000 mL of distilled water, followed by autoclaving. The homogenization of the sample in saline was followed by dilution of the homogenate in saline solution and then the inoculum was mixed well. After solidification, the Petri dishes were placed in a thermostat for cultivation. The product was microbiologically tested in PCA (plate count agar) (Penta; Prague, Czech Republic) for 3 days at 30 °C for determination of the total viable count (TVC) and MRS (Man Rogosa Sharpe agar) (Penta; Prague, Czech Republic) for 5 days at 30 °C for determination of the lactic acid bacteria (LAB) count. The results were recalculated as log_10_ CFU/g.

### 2.6. TBARS

The level of oxidation of the meat products during storage was determined by the formation of reactive thiobarbituric acid (TBARS) expressed as malondialdehyde (MDA) [25] content (mg/kg) measured at 538 nm. The homogenized sample (10 g) was transferred to a distillation tube containing 2.5 mL HCl (Penta; Prague, Czech Republic) diluted with distilled water (*v*:*v*; 1:2; HCl:H_2_O) and 97.5 mL of distilled water. The distillation tube was then placed in a BÜCHI K-355 distillation unit (BÜCHI; Flawil, Switzerland) and at 30% steam level the sample was distilled for 10 min until approximately 50 mL of distillate was collected. An aliquot of the distillate (5 mL) was then added to the boiling tube with 5 mL of 0.02 M 2-thiobarbituric acid (Merck KGaA; Darmstadt, Germany) (in a solution 90% acetic acid (Penta; Prague, Czech Republic) solution with distilled water) and the tube was boiled in a boiling water bath for 35 min. The absorbance of the complex was then measured at a given wavelength on an Onda V-10-plus (Giorgio Bormax S.r.l.; Carpi, Italy) spectrophotometer. The concentration of MDA was calculated according to the equation:(1)$$c_{MDA}(mg/kg)=7.8\times A\times \frac{m_{V}\times m_{D}}{10\times 50}$$
where *A* is the measured absorbance of the sample, *m_V_* is the weight of the sample (g), *m_D_* is the weight of the distillate (g), 10 is the theoretical weight of the sample (g), and 50 is the theoretical weight of the distillate (g).

### 2.7. Statistical Analysis

The statistical significance of the use of LP to reduce the concentration of malondialdehyde (n = 5) in the products was calculated using STATISTICA 12.0 CZ software (StatSoft; Prague; Czech Republic). Before statistical evaluation, a Dien–Dixon test was performed to exclude outliers. Mixed-design ANOVA was applied to the collected data. The product storage days were treated as a fixed effect and the MDA concentrations were included in the model as a random effect (MIXED procedure). An HSD Tukey test was then used to compare the mean values between samples. Microorganism counts were transformed into log_10_ CFU/g before determining means and performing statistical analyses by ANOVA. All microbiological analyses were conducted in triplicate (n = 3). Statistical significance was defined as *p* < 0.05 in all analyses.

## 3. Results

### 3.1. Effect of LP in Liquid Broth

The first part of this work was to observe the effect of LP on the growth of selected bacteria in liquid broths. The results are expressed as growth curves of the microorganisms, which were generated from the dependence of the optical density of the microorganisms on the cultivation time. Growth curves can be seen in Figure 1 and Figure 2.

The testing was carried out at the different temperatures 15, 20, 25, and 40 °C. Figure 1 presents the results of the growth curves of *L. innocua* in TSB broth. It can be seen that LP has inhibitory effects, especially at lower temperatures. A significant inhibitory effect (*p* < 0.05) was observed at the culture temperatures of 15 and 20 °C, when the lag phase of the bacteria was prolonged for approximately 25 h. On the other hand, at higher temperatures (30 and 40 °C) bacterial growth occurred more rapidly and even at 40 °C a higher optical density was detected. Inhibitory effects at a lower temperature can also be seen in the growth curve of *S. saprophyticus* (Figure 2A) and *P. fluorescens* (Figure 2B). The lag phase of *S. saprophyticus* was prolonged by approximately 10 h at 15 and 20 °C. The lag phase of *P. fluorescens* was not prolonged; however, a slower increase in the exponential phase can be seen.

### 3.2. Effect of LP on the Surface of Pork Cubes

At a storage temperature of 15 °C, the inhibitory effect was examined in untreated pork cubes, cubes inoculated with a solution of bacteria, and cubes inoculated with bacteria together with LP (5 g/100 mL). For this experiment, we used only *L. innocua* due to the most significant inhibitory effect of LP on these bacteria in a liquid broth experiment. Microbiological analyses on PCA agar were performed on days 1, 4, and 7 (see Table 1). Rapid growth of viable aerobic bacteria on PCA agar was observed in all pork cubes at 15 °C. Strong inhibition (*p <* 0.05) on the treated pork cubes using LP was obvious immediately during the first day of cultivation compared to control and control + *L. innocua* samples. The most significant difference (*p <* 0.05) in viable counts was on day 4. The difference between the LP samples and the others was between 0.8 and 1.3 log CFU/g. On day 7, the total counts in all samples were already between 8.8 and 9.8 log and the LP sample proved the inhibition of the number of bacteria present.

From these results, including an in vitro liquid broth experiment, the authors hypothesize the possible use of LP for the preservation of meat products. It should also be mentioned that the contamination at the beginning of the experiment was at a high level. The authors attribute this to contamination of the meat purchased. The control samples were not treated in any way.

### 3.3. Effect of LP in Pork Ham and Pâté

Due to the positive in vitro and ex vivo results, the number of microflora present in the selected meat products was also monitored. In addition to the number of microorganisms, the oxidative stability of meat products was also tested; this is closely related to the oxidative processes during the rendering process [22]. The results in Table 2 show what effect the addition of LP had on the development of microorganisms in pork ham and pâté. It should be noted that no lactic acid bacteria were found in the pork pâté product. The authors explain this phenomenon by a combination of a higher heat treatment temperature than that of ham and the addition of a seasoning mixture containing, for example, dried onions and garlic.

Table 2 shows the positive effect of LP on microbial contamination in both ham and pâté especially for the higher addition (0.5%) of LP2. A similar effect was also observed for the oxidation grade of the products, which showed lower levels of rancidity with the addition of LP1 and LP2, respectively. Slower growth of viable counts was found for both meat products. The higher concentration of LP2 inhibited bacterial growth more intensively compared to LP1, as expected. Throughout, the pork ham LP2 reduced all viable counts by approximately 0.5 log CFU/g (*p* < 0.05). Even more pronounced inhibitory effects were observed during the storage of pork pâté. We can observe up to 1 log CFU/g reduced viable counts for LP2 pâté. Table 1 shows that lactic acid bacteria accounted for the majority of the total viable counts. LP, even in this experiment, as we can see, did not completely stop the growth of bacteria, but inhibited and prolonged it. The results thus suggest that the shelf life of the products could be extended by up to 7 days.

The level of oxidation of meat products is shown in Figure 3. Similar results to the microbiological stability were confirmed for the oxidation level. Meat products with a higher concentration of LP2 showed less rancidity as they contained a lower malondialdehyde content (mg/kg). The MDA content of the sample was always lower with an increasing addition of LP. The reduction of MDA in LP2 was significant during all tests. On day 7, the MDA content of LP2 ham was almost half that of the control (*p* < 0.05), the same as on day 28. The MDA content of LP1 was also lower, although not necessarily significant. The MDA content of the ham at the beginning of the experiment was close to zero mg/kg. On the other hand, on the first day of the experiment for pork pâté, oxidation was observed due to the higher fat content. A more significant slowing down of oxidation processes is evident in the pork pâté. The MDA content of LP2 pâté was in most cases approximately three times lower during each day of the experiment compared to the control sample. There was also a significant reduction in LP1, about two times lower. In general, an MAD concentration of more than 2 mg/kg already causes a negative sensory flavor in products, a typical rancid aroma and a pungent and bitter taste. The level of 2 mg/kg for ham was not reached, while the control sample of pâté rigidly exceeded the level already on day 7. Therefore, the shelf life of the products can be extended again because of these findings on oxidation processes.

## 4. Discussion

The microbial stability of meat products depends on many factors related to the whole process of meat production and the manufacture of meat products, in particular, the temperature, which has the greatest effect on the potential of microbes to reduce the quality and shelf life of products. In addition, technologists are also faced with an increasing customer demand for safe additives that do not affect the quality of the final product and human health. LP as a substitute for chemical preservatives could have good potential to ensure the microbial safety of meat products [9,26]. Most often, we can find professional publications dealing with the use of LP in the dairy industry, especially as a preservative in developing countries where the rapid cooling of milk is difficult [16,27,28,29].

LP shows a strong dependence on the temperature at which it is applied. From the findings of the temperature dependence of LP activity, it was found that the highest LP activity increases with a decreasing temperature to 4 °C of milk [16]. The results of our study are consistent with this claim, since according to Figure 1 and Figure 2. We can see that inhibitory activity against *L. innocua*, *S. saprophyticus*, and *P. fluorescens* bacteria was highest at the tested temperature of 15 and 20 °C. Therefore, it was found that the lag phase of *L. innocua* was prolonged up to two times. Vice versa, at higher temperatures, the mentioned bacteria had a shortened lag phase. A similar trend was observed in a study looking at the inhibition of foodborne bacteria in the beef cube system [9] and the inhibitory effect on *E. coli* 0157:H7, *L. monocytogenes* and *S. aureus* [1].

Meat and meat products can be infected by a wide range of microorganisms and, given the confirmed inhibitory effects of LP on Gram-negative bacteria and some Gram-positive bacteria in milk and dairy products [12,13,14,15,16], a similar effect can be expected in meat and meat products. There are not many studies dealing with LP in meat as far as the authors are aware. For example, the inhibitory effect of LP on the growth of *E. coli* O157:H7, *L. monocytogenes* L45 and *S. aureus* R37 in a 37 °C broth system and in minced red meat (0, 6 and 12 °C) was investigated [1]. *L. monocytogenes* was shown to be the bacterium most susceptible to LP, and after a 4 h culture period, LP significantly reduced the growth of microbial populations. A similar study focused on the surface microflora of beef meat cubes. The cubes were inoculated with approximately 10^4^ CFU/cm of bacteria and treated with LP. Incubation was carried out under different conditions (24 h at 37 °C, 7 days at 12 °C, 7 days at 12 to −1 °C, 4 weeks at −1 °C) [9]. Inhibition by LP was effective at temperatures suboptimal for the growth of the chosen bacteria. Strong inhibition in cubes was found at 12 °C as well as a reduction in viable pathogens at refrigeration temperatures (especially *Pseudomonas* spp.). However, there was no arrest in the development of native lactic acid bacteria. The results of our work investigating LP influences on the surface of pork cubes agree with previous studies [1,9]. Pork cubes inoculated with *L. innocua* had 1 log CFU/g of higher viable findings at the end of the storage trial (day 7 at 15 °C). Different concentrations (0, 3, 5, and 7% *w*/*w*) of LP were also used in whey protein isolate to treat roast turkey [30]. The results showed a reduction of 3 and 2 log CFU/g in inoculated samples of *S. enterica* and *E. coli* O157: H7, respectively. The inhibitory effect was confirmed on both bacteria at storage temperatures of 4 and 10 °C. Thus, the results of our study and the previous studies mentioned suggest the possibility of prolonging the shelf life of meat and meat products using LP, either in the form of a film on the surface or mixed into the mixture during manufacture.

One of the more recent studies investigated the effect of different types of active biodegradable films containing LP on the shelf life of fish burgers [31]. Fish products were stored at 4 °C for 20 days. The study confirmed lower total viable counts for products wrapped in LP + albumin and LP + chitosan films at day 1. At the end of the experiment, there was a reduction in viable counts of approximately 3 log. In contrast, on day 1, the total counts were higher for products wrapped only in films containing albumin or chitosan compared to biodegradable films containing LP only. Similar results were found for the growth of psychrophilic microorganisms and *Pseudomonas* spp. and *Shewanella* spp. Our results for pork meat products are therefore consistent with the results of this study. However, the differences in TVC in our study were not as marked (see Table 2) but it is worth noting that LP2 showed its inhibitory efficacy especially in pork pâté (*p* < 0.05). Differences in LP efficacy between studies may have been due to differences in methodology such as wrapped fish products in LP film. In our study, LP was added to the product during manufacturing. Another factor that may have caused the difference is the increase in antimicrobial activity of the LP solution incubated at 23 ± 2 °C for 24 h [31].

The rate of lipid oxidation in the products was determined using a thiobarbituric acid test, in which the MDA content was measured. MDA is formed as a secondary oxidation product from hydroperoxides of the lipids present in the products [32]. Lipids are susceptible to degradation and are one of the main non-microbial reasons for the reduced quality of meat and meat products [30]. The effect of LP on oxidation rates has been described in several publications. It was also found that the combination of the lactoperoxidase system and the whey protein used as shrimp coating increased TBARS during storage at 4 °C [33]. Similar results have already been published showing no significant changes in MDA formation in samples of rainbow trout and pikeperch fillets treated with LP [34,35,36]. Furthermore, reduced MDA levels were also confirmed when shrimps were placed in cold storage. However, the reason for the reduced oxidation rate was not directly due to LP but to the biodegradable film containing LP that caused a barrier of meat to oxygen and light [32]. Therefore, our work does not confirm or agree with these studies. In fact, from the results of our work, we can see (Figure 3) a significantly lower level of MDA (*p* < 0.05) in products containing LP, especially at a concentration of 0.50%. Our hypothesis is of reduced MDA production due to lower concentrations of microorganisms (Table 2) as a result of the inhibitory effect of LP on bacteria. In fact, bacteria are capable of lipid peroxidation products in cell membranes, including malondialdehyde [37,38]. Therefore, from our results we can say that LP extends the shelf life of meat products.

Evaluating the impact of LP to preserve meat and meat products and other commodities is just at the beginning of testing and observation. As already mentioned, there is a lack of studies on the subject as far as the authors are aware. Naturally occurring LP is and has often been tested for preserving milk of various types. The authors are aware of some of the shortcomings within the study. In future experiments, we would like to focus on other types of meat products as well as other commodities. Certainly, the authors consider it worthwhile in future studies to look at the effect of LP on beneficial microorganisms such as lactic acid bacteria. These bacteria are used for their positive fermentation processes in the maturation of meat products. It would also be interesting to test the effect of LP on, for example, dried meat products (jerky) in view of the possible risk of mold growth on the surface of the product. However, the authors do not intend to diminish the quality of this paper, but rather to motivate further research.

## 5. Conclusions

Currently, there is a lack of scientific publications that focus on the preservation of meat and meat products by lactoperoxidase. From the results of this work, it is possible to note the positive effect of lactoperoxidase on inhibition of the growth of *L. innocua*, *S. saprophyticus*, and *P. fluorescens*. The effectiveness is limited by temperature, being most effective at temperatures that are not optimal for bacterial growth. This fact was exploited for the storage of pork hams and patties at 4 °C. There was a reduction in viable counts without affecting lactic acid bacteria. Another positive finding was a reduction in malondialdehyde formation as a product of the lipid secondary oxidation of lipids. Thus, in addition to its use in the dairy industry, there is also the potential to use lactoperoxidase to preserve meat or meat products.

## Figures and Tables

**Figure 1 microorganisms-12-01010-f001:**
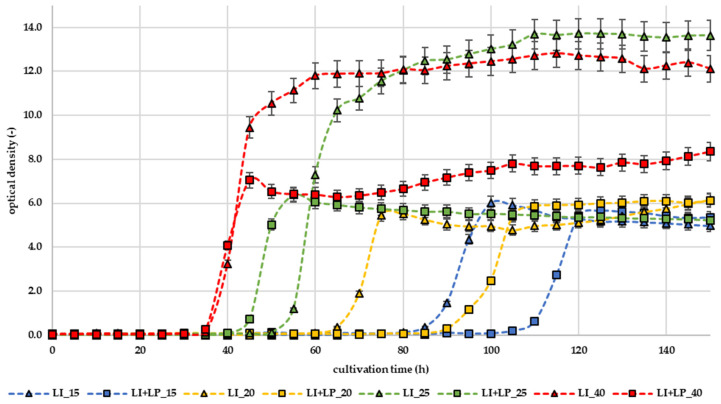
Effect of LP on *L. innocua* growth curves; LI (∆)—control bacteria; LI + LP (□)—bacteria + lactoperoxidase; 15, 20, 25, and 40—temperatures of cultivation (°C).

**Figure 2 microorganisms-12-01010-f002:**
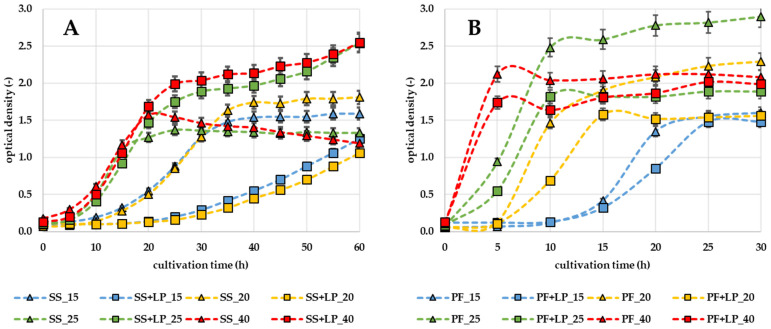
Effect of LP on *S. saprophyticus* (**A**) (SS) and *P. fluorescens* (**B**) (PF) growth curves; SS or PF (∆)—control bacteria; SS + LP or PF + LP (□)—bacteria + lactoperoxidase; 15, 20, 25, and 40—temperatures of cultivation (°C).

**Figure 3 microorganisms-12-01010-f003:**
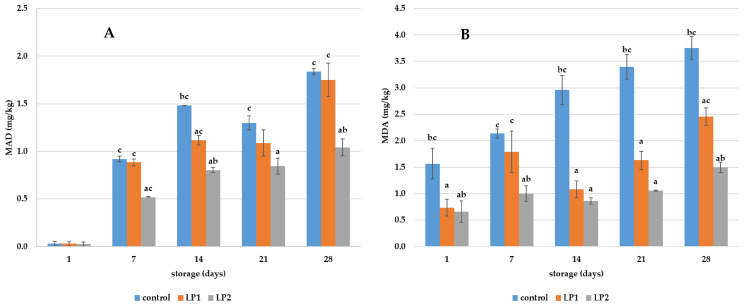
Oxidation level (MDA concentration (mg/kg) of meat products ((**A**)—ham, (**B**)—pâté) with/without the addition of LP; LP1—meat products with 0.25% of LP; LP2—meat products with 0.50% LP; a–c means statistical difference (*p* < 0.05) between control and LP1, respectively LP2.

**Table 1 microorganisms-12-01010-t001:** TVC at days 1, 4, and 7 at 15 °C of pork cubes soaked in distilled water (control), cubes soaked in bacteria inoculum (*L. innocua*), and soaked in inoculum and LP solution (*L. innocua* + LP).

Treatment	Total Viable Count (log_10_ CFU/g ± SD) ^1^		
	Day 1	Day 4	Day 7
control	5.2 ^c^ ± 0.1	7.7 ^bc^ ± 0.2	9.1 ^bc^ ± 0.2
*L. innocua*	5.3 ^c^ ± 0.1	8.2 ^ac^ ± 0.1	9.8 ^ac^ ± 0.0
*L. innocua* + LP	4.7 ^ab^ ± 0.1	6.9 ^ab^ ± 0.1	8.8 ^ab^ ± 0.1

^1^ values presented are the mean of three replicates ± standard deviation; means followed by the different lowercase letter in the column did differ significantly (*p* < 0.05).

**Table 2 microorganisms-12-01010-t002:** Microbiological analysis of meat products with/without the addition of LP.

Meat Product	Storage (Days)	Count (log_10_ CFU/g ± SD ^a^)					
		TVC			LAB		
		Control	LP1	LP2	Control	LP1	LP2
Pork ham	1	4.0 ^bc^ ± 0.1	3.7 ^ac^ ± 0.1	3.4 ^ab^ ± 0.1	2.7 ^bc^ ± 0.1	2.4 ^a^ ± 0.0	2.5 ^a^ ± 0.1
	7	4.5 ^c^ ± 0.1	4.4 ^c^ ± 0.0	3.9 ^ab^ ± 0.2	4.0 ^c^ ± 0.2	3.8 ± 0.1	3.7 ^a^ ± 0.1
	14	5.0 ^c^ ± 0.2	4.5 ± 0.0	4.3 ^a^ ± 0.1	4.5 ^bc^ ± 0.2	3.8 ^a^ ± 0.0	3.8 ^a^ ± 0.1
	21	5.2 ^bc^ ± 0.1	5.1 ^a^ ± 0.1	4.9 ^a^ ± 0.1	5.1 ^c^ ± 0.1	4.9 ± 0.1	4.7 ^a^ ± 0.1
	28	6.3 ^bc^ ± 0.1	6.2 ^ac^ ± 0.1	5.9 ^ab^ ± 0.1	6.1 ^c^ ± 0.1	5.9 ± 0.1	5.8 ^a^ ± 0.0
Pork pâté	1	4.2 ^c^ ± 0.2	4.0 ± 0.1	3.8 ^a^ ± 0.1	<1.0	<1.0	<1.0
	7	5.0 ^c^ ± 0.0	4.8 ^c^ ± 0.2	4.5 ^a^ ± 0.1	<1.0	<1.0	<1.0
	14	5.9 ^bc^ ± 0.1	5.4 ^ac^ ± 0.0	4.7 ^ab^ ± 0.2	<1.0	<1.0	<1.0
	21	6.3 ^c^ ± 0.2	6.2 ± 0.1	5.1 ^a^ ± 0.0	<1.0	<1.0	<1.0
	28	6.6 ^bc^ ± 0.1	6.1 ^ac^ ± 0.1	5.6 ^ab^ ± 0.0	<1.0	<1.0	<1.0

^a^ values presented are the mean of three replicates ± standard deviation; means followed by the different lowercase letter in the row did differ significantly (*p* < 0.05); TVC—total viable count; LAB—lactic acid bacteria.

## Data Availability

The data presented in this study are available on request from the corresponding author. The data are not publicly available due to privacy reasons.

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
