# Peer review of "Use of Lactoperoxidase Inhibitory Effects to Extend the Shelf Life of Meat and Meat Products"

_microorganisms, 2024, doi:10.3390/microorganisms12051010_

Round 1

Reviewer 1 Report

Comments and Suggestions for Authors

microorganisms-3013558-peer-review-v1

Paper is interesting, however, on the way that experiments were performed and presented results is very difficult to proof that suggested approach will really work for the bio conservation of meat. In my opinion, some extensive revision of the paper, help from more experience colleagues can be option to present better the manuscript. Moreover, some additional experiments, and some more appropriate controls, maybe model system, and not really real meat samples can be option to convince readers that application of LP can be way for improving food safety of meat products. Provide visual material (Figures and table need to be presented better and appropriate legends provided). In conclusion, better presentation, better structure, and some additional experiments are needed in order for the paper to be suggested for publication.

Ln43-45: the sentence is not clear. Please, check and correct.

Ln43: Gram need to be with capital G. Please, check the entire manuscript and adjust accordingly.

Ln42-50: This section is presented in very general way. Maybe authors can consider adding a bit more about mode of action of LP. Including interaction with specific amino acids and interaction with microbial stability.

Please, after introducing specific bacterial species, in following occasions, name need to be abbreviated according to the recommendations form the journal.

Authors will need to choose examples where pathogen/spoilage microbes will be mentioned and also to discuss if the LP can affect beneficial microbes. But please, do not mix spoilage/pathogens and beneficial in same examples.

Ln62: You have stated that LP is thermo stable, however, on Ln56 you have mentioned that "30 seconds when milk is heated to 80 °C.". This is a bit of a controversial statement (mentioned on Ln 62) since you have provided different information before. Please, be more fact strait.

Ln58, 68, etc.: reference need to be according to the instructions from journal.

In my opinion Introduction can be presented better, with a bit more focus on mechanisms of action of LP and provide some more examples. Maybe help from more experienced colleagues can be a good option in improving the structure of the manuscript.

Ln83: Mixture of what enzymes? Please be more specific. These enzymes were obtained from what origine?

Ln87-104, 106-116: The applied procedure was not well described.

Any controls were applied? Please, explain all details of the experimental approaches.

In order to observe killing effect, a cell count, and CFU/ml will be more appropriate than OD monitoring.

Is OD of 14 realistic?

Material and methods description do not match with results. Some of the results are not link to well described material and methods.

Table 1: looks like control is highly contaminated. Do you know what kind of bacteria are part of that natural contaminations? Well, a reduction of 1 log is not really sufficient to claim that LP was effective in providing extension pf the shelf life.

References need additional attention. Most of them missing journal volumes, do not need to say "pp" and names need to be separated by ";".  Journal names need to be abbreviated. Please, check instructions for authors and some of recently published papers and use it as model.

Author Response

Dear reviewere No. 1.

Thank you for your time spent on our Manuscript. We hope that we have completed, corrected and, explained everything to your liking. 

We are uploading an MS Word file with the responses to your suggestions. 

Reviewer 2 Report

Comments and Suggestions for Authors

The manuscript was interesting and validated the use of lactoperoxidase in extending the shelf life of meat products. The paper was also well written but had the following problems:

1. The authors need to explain why they used 3 microorganisms in the growth curve culture and why they used 2 in the subsequent experiments?

2. The authors need to point out the shortcomings of this study and look ahead to what needs to be further explored in the future.

Author Response

Dear reviewere No. 1.

Thank you for your time spent on our Manuscript. We hope that we have completed, corrected and explained everything to your liking. 
We appreciate your comments on our Manuscript.

We are uploading an MS Word file with the responses to your suggestions. 

Round 2

Reviewer 1 Report

Comments and Suggestions for Authors

Authors have improved the text, however, still not fully convinced that presented work have real application for the food industry.